# Amyloid β-Peptide Increases Mitochondria-Endoplasmic Reticulum Contact Altering Mitochondrial Function and Autophagosome Formation in Alzheimer’s Disease-Related Models

**DOI:** 10.3390/cells9122552

**Published:** 2020-11-28

**Authors:** Nuno Santos Leal, Giacomo Dentoni, Bernadette Schreiner, Luana Naia, Antonio Piras, Caroline Graff, Antonio Cattaneo, Giovanni Meli, Maho Hamasaki, Per Nilsson, Maria Ankarcrona

**Affiliations:** 1Division of Neurogeriatrics, Department of Neurobiology, Care Science and Society, Karolinska Institutet, BioClinicum J9:20, Visionsgatan 4, 171 64 Solna, Sweden; giacomo.dentoni@ki.se (G.D.); bernadette_schreiner@gmx.de (B.S.); luana.naia@ki.se (L.N.); Antonio.piras.job@gmail.com (A.P.); caroline.graff@ki.se (C.G.); per.et.nilsson@ki.se (P.N.); 2European Brain Research Institute (EBRI), Viale Regina Elena 295, 00161 Roma, Italy; a.cattaneo@ebri.it (A.C.); g.meli@ebri.it (G.M.); 3Department of Genetics, Graduate School of Medicine, Osaka University, 2-2 Yamadaoka, Suita, Osaka 565-0871, Japan; hamasaki@fbs.osaka-u.ac.jp

**Keywords:** Alzheimer’s disease, amyloid β-peptide, autophagy, Mitochondria-ER contact sites, mitochondria

## Abstract

Recent findings have shown that the connectivity and crosstalk between mitochondria and the endoplasmic reticulum (ER) at mitochondria–ER contact sites (MERCS) are altered in Alzheimer’s disease (AD) and in AD-related models. MERCS have been related to the initial steps of autophagosome formation as well as regulation of mitochondrial function. Here, the interplay between MERCS, mitochondria ultrastructure and function and autophagy were evaluated in different AD animal models with increased levels of Aβ as well as in primary neurons derived from these animals. We start by showing that the levels of Mitofusin 1, Mitofusin 2 and mitochondrial import receptor subunit TOM70 are decreased in post-mortem brain tissue derived from familial AD. We also show that Aβ increases the juxtaposition between ER and mitochondria both in adult brain of different AD mouse models as well as in primary cultures derived from these animals. In addition, the connectivity between ER and mitochondria are also increased in wild-type neurons exposed to Aβ. This alteration in MERCS affects autophagosome formation, mitochondrial function and ATP formation during starvation. Interestingly, the increment in ER–mitochondria connectivity occurs simultaneously with an increase in mitochondrial activity and is followed by upregulation of autophagosome formation in a clear chronological sequence of events. In summary, we report that Aβ can affect cell homeostasis by modulating MERCS and, consequently, altering mitochondrial activity and autophagosome formation. Our data suggests that MERCS is a potential target for drug discovery in AD.

## 1. Introduction

Alzheimer’s disease (AD) is the most common form of dementia and leads to cognitive impairment, loss of memory, changes in behavior, decreased ability to accomplish activities of daily living and, ultimately, the need for full-time care [1]. Two major molecular hallmarks of this disease are extracellular amyloid plaques, constituted of amyloid β-peptide (Aβ), and intracellular neurofibrillary tangles, constituted of hyperphosphorylated tau protein [1]. Aβ is formed by the subsequent cleavage of amyloid precursor protein (APP) by β- and γ-secretases [1]. Recently, we and others have shown that all components required for APP processing are present in mitochondria-endoplasmic reticulum (ER) contact sites (MERCS) and that formation of Aβ takes place in this specialized subcellular region [2,3,4]. Furthermore, we demonstrated that modulation of MERCS leads to changes in Aβ levels [5] and we also reported a positive correlation between the number of MERCS and ventricular levels of Aβ42, which is believed to be the most toxic species of Aβ [6,7], in idiopathic normal pressure hydrocephalus patient’s brain biopsies obtained during neurosurgery [8].

MERCS are subcellular domains arising from the interaction between specific sub-regions of the ER and the outer membrane of mitochondria. At these interorganellar contact sites, essential biological processes occur, such as autophagosome formation and calcium (Ca^2+^) shuttling from ER to mitochondria [9,10]. Interestingly, MERCS ultrastructure and function have been demonstrated to be increased in different AD models, although the causes of these alterations remain to be fully understood [8,11,12,13]. So far, over 75 proteins have been reported to localize at MERCS [14]. While some of these proteins predominantly have biological functions in this subcellular region (e.g., translocase of the outer mitochondrial membrane 70 (TOM70), inositol trisphosphates receptor type 3 (IP3R3) and voltage-dependent anion-selective channel 1 (VDAC1)), others are suggested to have primarily structural roles maintaining the connectivity between the two organelles (e.g., mitofusin 1/2 (Mfn1/2), vesicle-associated membrane protein-associated protein B (VAPB) and protein tyrosine phosphatase interacting protein 51 (PTPIP51)) [15,16,17,18,19].

Macroautophagy, herein referred to as autophagy, is a cellular process that is responsible for delivering cytosolic components and organelles to the lysosome to be degraded and recycled. In 2005, it was reported that immature autophagic structures—called autophagic vacuoles (AVs)—accumulate in the brain of AD patients, suggesting that autophagy homeostasis is altered in the disease [20]. More recently, autophagosomes have been reported to have a role in the production, secretion and degradation of Aβ, suggesting their potential role in the etiology, pathology and/or progression of AD [20,21,22]. Although the exact mechanisms regulating the formation of the isolation membrane or autophagosomes are not fully understood, it has been reported that MERCS are one of the places where autophagosome formation can occur in the cell [23,24,25]. Indeed, Hamasaki and colleagues showed that knock-down of Mfn2 leads to reduction of the pool of autophagosomes, and Gomez-Suaga and colleagues showed similar results when increasing MERCS by overexpression of VAPB and PTPIP51 [23,26]. Adenosine triphosphate (ATP) is one of the molecules which has been described to modulate autophagosome biogenesis, either by being essential for autophagosome component formation or by activating the 5′ adenosine monophosphate (AMP)-activated protein kinase (AMPK) signaling pathway [27,28,29,30]. In fact, acute amino acid starvation results in increased ATP levels and mitochondrial function [30,31]. Since Ca^2+^ transfer from ER to mitochondria has been reported to influence mitochondrial function and ATP production [32], these data suggest that MERCS alterations in AD could influence dysregulation of mitochondrial function and autophagosome synthesis observed in this pathology.

Here, we show that post-mortem familial AD (FAD) patients’ brain present alterations in the MERCS-related proteins Mfn1/2 and TOM70. Moreover, we demonstrate for the first time that the connectivity between ER and mitochondria is increased in brain tissue and primary neurons obtained from AD mouse models displaying increased Aβ levels. We also show that MERCS are altered at different time points during the pathology progression in these AD mouse models. These ultrastructural alterations are in concomitance with alterations in autophagosome formation, mitochondrial membrane potential, oxygen consumption and ATP production during starvation. In summary, we present a new mechanism suggesting that Aβ affects cell homeostasis by increasing mitochondria–ER apposition, subsequently altering mitochondrial function and autophagosome biogenesis.

## 2. Material and Methods 

### 2.1. Antibodies

The following antibodies were used: β3 tubulin (Santa Cruz Biotechnology, Dallas, TX, USA, #80016), Actin (Sigma-Aldrich, St. Louis, Missouri, USA, #A4700), APP 6E10 (BioLegend, San Diego, CA, USA, #803001), APP Y188 (Abcam, Cambridge, UK, #ab32136), Drp1 (BD Bioscience, San Jose, CA, USA, #611112), GAPDH (Enzo LifeScience, Exeter, UK, #ADI-CSA-335-E), IP3R3 (BD Biosciences, San Jose, CA, USA, #610312), LC3B (Cell Signalling, Danvers, MA, USA, #3868; Novus Biologicals, Centennial, CO, EUA, #NB100-2220), Mfn1 (Santa Cruz Biotechnology, #SC50300), Mfn2 (Abcam, #Ab56889), Opa1 (BD Bioscience, San Diego, CA, USA, #612606), SQSTM1/p62 (Cell Signalling,#5114), TIM23 (BD Biosciences, #611223), TOM20 (Santa Cruz Biotechnology, #sc-11415), TOM70 (Santa Cruz Biotechnology, #sc-366282) and VDAC1 (Abcam, #Ab14734).

### 2.2. Human Post-Mortem Brain Samples Preparation 

Post-mortem brain samples from the frontal cortex from FAD *APP^Swe^* (*n* = 4, ages between 56 and 66, post-mortem time between 24 and 40 h) and non-demented controls (*n* = 4, ages between 67 and 82, post-mortem time between 9 and 27 h) were used to assess MERCS-related proteins (further description of patients in Appendix A). Frozen tissue was cut in pieces while kept cold on ice and homogenized with 5 cm^3^ glass-Teflon homogenizer (1200 rpm) in digestion solution (150 mM NaCl, 1% Triton X-100, 0.5% sodium deoxycholate (DOC), 0.1% sodium dodecyl sulfate (SDS), 50 mM Tris pH = 7.5 (all chemicals from Sigma-Aldrich), 1x mammalian ProteaseArrest (G-Biosciences, St. Louis, MO, USA #786-433), Phosphatase Inhibitor Cocktail 3 (Sigma-Aldrich, #P0044) and 1x Benzonase (Merck, Darmstadt, Germany, #70664)). After 10 min of incubation, a quick homogenization was performed with the same Teflon homogenizer (1200 rpm) and the homogenate centrifuged at 7000× *g*, for 10 min at 4 °C to remove debris. Supernatant was collected and protein concentration determined by the Pierce™ bicinchoninic acid assay BCA Protein Assay Kit (ThermoFisher, #23225). All the steps were performed on ice.

### 2.3. Animal Models

In order to establish a correlation between MERCS ultrastructure and disease progression, wild-type (WT) mice and three different AD mouse models with the same genetic background (C57BL/6) were used: (i) transgenic (tg) mice overexpressing (OE) APP with the Swedish (K670N/M671L) and London (V717I) mutations (*App^Swe/Lon^*) [33], (ii) *App* knock-in (KI) mice with the Swedish and Iberian (I716F) mutations (*App^NL -F^*) and (iii) *App* knock-in mice with the Swedish, Iberian and Arctic (E693G) mutations (*App^NL-G-F^*) [34]. The disease progression in the three different animal models used are as follows: (i) *App^Swe/Lon^*-plaque formation in the neocortex at 4 months and in the hippocampus at 6 months, memory deficits starting from 6 months, (ii) *App^NL-F^-*plaque formation after 6 months in cortex and hippocampus, synaptic loss at 9 months and memory impairment after 18 months and (iii) *App^NL-G-F^*-plaque formation at 2 months in cortical and subcortical areas, synaptic loss at 4–5 months and memory impairment at 6 months [33,34]. Tissue from the *App*^Swe/Lon^ mice were a kind gift from QPS Austria. Animals were generated, housed, and sacrificed on conformity to the Austrian guidelines for the care and use of laboratory animals. *App^NL -F^* and *App^NL-G -F^* animal models were a kind gift of Dr Takaomi Saido and Dr Takashi Saito, RIKEN Center for Brain Science, Tokyo, Japan. All stated mutations have been identified in FAD patients. In general, these mutations lead to increased production of Aβ with an increased Aβ42/Aβ40 ratio, except the Arctic mutation (located within the Aβ sequence), which increases Aβ protofibril formation. The animals were housed in a 12:12 h light/dark cycle and with ad libitum access to food and water. Hippocampus *Cornu Ammonis* area 1 (CA1) and cortex from animals of four different ages 2, 4, 6.5 and 10 months (WT and *App^Swe/Lon^*) and three different ages 4, 6.5 and 10 months (*App^NL-F^* and *App^NL-G-F^*), were analyzed (in total 48 animals). The ages of animals were chosen to be able to assess MERCS ultrastructure before, during and after plaque formation. In brief, anesthetized animals were directly perfused with 2% glutaraldehyde (LADD Research Industries, Williston, VT, USA, #20105) and 1% paraformaldehyde (Agar Scientific, Stansted, UK, #1026) in 0.1 M phosphate buffer and brains were dissected and hemisected. Right hemispheres were put in fixation solution overnight and the next day placed in a brain slicer matrix where hippocampi (CA1) and cortices were collected and processed for ultrathin sectioning for ultrastructural transmission electron microscopy (TEM).

### 2.4. Primary Cortical Neurons Preparation

Primary cortical neurons (PCN) (embryonic day 16-17) derived from WT or *App^NL-F^* mice were seeded in poly-D-lysine (Sigma-Aldrich, #P7405-5mg)-coated plates in Neurobasal medium (Gibco, Waltham, MA, EUA, #21103049) supplemented with 1x B-27 (ThermoFisher, #17504044) and 2 mM of l-Glutamine (ThermoFisher,, #25030081). At 7 days in vitro (DIV), half of the cell culture media was replaced with freshly prepared media. At 13–16 DIV, cells were either exposed to Aβ or starved and then harvested. Starvation is a well-established method to induce autophagy and was chosen here for this purpose [35]. To induce amino acid starvation, media was removed, and cells were washed with Earle’s Balanced Salt Solution (EBSS) (Gibco, Waltham, MA, EUA, #14155-063). Subsequently, cells were incubated in EBSS between 0.5 to 3 h. Bafilomycin A1 (Sigma-Aldrich, #B1793) (100 nM) treatment were done during the same respective starvation period (3 h for fed condition).

### 2.5. Aβ Treatment of Cell Cultures

Monomeric and oligomeric Aβ1-42 (mAβ42 and oAβ42, respectively) were prepared according to Fa and colleagues [36]. The grouping into mAβ42 and oAβ42 was done based on previous characterization of these species [37]. In brief, Aβ42 peptide (AnaSpec, Fremon, CA, #AS-24224) was dissolved in siliconized Eppendorf tubes in ice-cold hexafluoroisopropanol (Alfa Aesar, Kandel, Germany, #A12747) to obtain a solution of 1 mM which was left to evaporate overnight. The pellet was then resuspended in dimethyl sulfoxide (DMSO) (Sigma-Aldrich, # 276855) and sonicated in a water bath for 10 min to a working concentration of 5 mM and kept at −80 °C for up to 6 months. mAβ42 was prepared by diluting the 5 mM Aβ42 stock with DMSO to a concentration of 1 mM and then added directly to the cells. oAβ42 was prepared in the same way except that the solution was kept at 4 °C for 24 h before use, allowing the oligomerization. Both mAβ42 and oAβ42 were added to the media of the cells at a final concentration of 2 µM for 24 h. The antibody fragment scFvA13 (final concentration of 40 nM), purified as recombinant protein, was added at the same time as Aβ and it was used as a control to selectively target oAβ42 and silence its function [38,39].

### 2.6. Transmission Electron Microscopy (TEM) Analysis

Ultrathin sections were prepared using Leica Ultracut UCT (Leica, Vienna, Austria) and contrasted with uranyl acetate and lead citrate. Sections were observed in a Tecnai 12 BioTWIN (FEI Company, Eindhoven, The Netherlands) or JEM-1011 (JEOL, Inc., Peabody, MA, USA) transmission electron microscope at 100 kV. Digital images were taken with a Veleta camera (Olympus Soft Imaging Solutions, GmbH, Münster, Germany) or 2k × 2k advanced microscopy techniques (AMT) mid-mount digital camera (Advanced Microscopy Techniques, Corp. Woburn, MA, USA) at a primary magnification of 26,500× or 30,000×.

For the animal models’ pictures, all mitochondria of 10 different cells were snapped per brain region (CA1 and cortex) per animal (WT, *App^Swe/Lon^*, *App^NL-F^* and *App^NL-G-F^*) per age (2, 4, 6.5 and 10 months) (total of 280 cells). Similarly, for PCN pictures, all mitochondria of 10 different cells per condition were snapped. A MERCS was considered when the distance between ER and mitochondria was ≤30 nm. Number of MERCS and number of mitochondria profiles were counted while MERCS length (length of apposition between ER and mitochondria where the distance between the organelles is ≤30 nm) and mitochondria profile perimeter were quantified using the freehand line tool in ImageJ (NIH, USA). Number of MERCS per mitochondria was obtained by dividing the number of MERCS per number of mitochondria profile and % mitochondria surface in contact with ER obtained by dividing MERCS length by mitochondrial perimeter and multiplied by 100. For mouse brain tissue, each dot represents the average of 4 randomly selected pictures per animal followed by the average of each animal type (*n* = 3 (WT and *App^Swe/Lon^*) or *n* = 4 (*App^NL-F^* and *App^NL-G-F^*)), while in PCN, each dot corresponds to the average of each cell.

### 2.7. Cell Lysis and Western Blot

PCN were lysed in digestion solution with proteinase and phosphatases inhibitors, and benzonase solution (50 mM Tris pH = 8.0, 4 mM MgSO_4_ and 1x benzonase - Sigma-Aldrich). As before, protein concentration was measured using the Pierce™ BCA Protein Assay Kit.

10–25 µg of protein was loaded and run on 4–12% Bis-Tris gel (or 12% for LC3B) (Novex, ThermoFisher, Waltham, MA, USA) and transferred to nitrocellulose (or methanol-activated polyvinylidene fluoride for LC3B) membrane. Membranes were blocked in 5% milk Tris-buffered saline with tween (TBS-T), incubated overnight at 4 °C with first antibody, washed in TBS-T, incubated for 1 h at room temperature with secondary antibody in 5% milk and finally washed in TBS-T. Membranes were developed using Odyssey CLx (LI-COR, Cambridge, UK), and quantifications were performed using ImageJ or Image Studio Lite (LI-COR, Cambridge, UK), normalized to respective loading control (GAPDH, actin or tubulin) and, for starved conditions, normalized to respective fed condition.

### 2.8. Enzyme-Linked Immunosorbent Assay (ELISA)

PCN were grown in 12-well plates (400,000 cells per well) and, after 13 DIV, media was adjusted to 1 mL and incubated with 10 µM γ-secretase inhibitor L-685,458 (Sigma-Aldrich, #L1790-5MG). After 24 h, media was removed, protein concentration determined and Aβ42 concentration was measured using Amyloid*β* (1–40) and (1–42) Assay Kit-IBL (IBL, Minneapolis, MN, #16340 and #16233, respectively), according to the manufacturer’s instructions.

### 2.9. Seahorse Analysis

WT- and *App^NL-F^*-derived PCN were grown in Seahorse XF96 microplate (Agilent, Santa Clara, CA, USA) (26,000 cells per well) for 14 DIV. At 15 DIV, cells were starved with EBSS as described above. During the last 30 min, starved cells were calibrated in EBSS media without sodium bicarbonate (KCl 5.3 mM, NaCl 117.24 mM, NaH_2_PO_4_ · H_2_O 1.01 mM, d-glucose 5.5 mM, pH = 7.4–all from Sigma-Aldrich), while fed cells were calibrated in Dulbecco’s Modified Eagle Medium (DMEM) media without sodium bicarbonate (DMEM (Sigma-Aldrich, #D5030), Sodium pyruvate 0.22 mM, l-glutamine 200 µM, d-glucose 25 mM, pH = 7.4-all from Sigma-Aldrich, St. Louis, Missouri, EUA), at 37 °C and without CO_2_. Basal respiration, ATP production and maximal respiration were measured using Seahorse XFe96 Analyzer (Agilent), according to the Seahorse XF Cell Mito Stress Test Kit (Agilent) and values obtained were normalized to amount of protein and respective fed condition. The final concentrations of the drugs used were: Oligomycin (Oligo) (1 µM), Carbonyl cyanide-4-(trifluoromethoxy) phenylhydrazone (FCCP) (1 µM) and Rotenone + Antimycin A (Rot + AA) (0.5 µM).

### 2.10. TMRM and ATP Levels

PCN were grown in 96-well plates (45,000 cells per well) for 14 DIV. At 15 DIV, cells were starved for 0.5 up to 3 h. 0.5 h before the end of the experiment, EBSS without sodium bicarbonate (as before) containing 150 nM (quenching condition) of tetramethylrhodamine, methyl ester (TMRM) (ThermoFisher, #T668) was added to the cells. After 25 min, TMRM basal fluorescence was measured and, after 5 min, FCCP (2.5 µM) and Oligomycin (3.2 µM) were added to the samples to induce TMRM release from mitochondria and fluorescence was measured (λ_Ex_ = 544 nm; λ_Em_ = 590 nm, 37 °C). For ATP levels, cells were seeded and starved in the same way and the CellTiter-Glo^®^ Luminescent Cell Viability Assay (Promega, Madison, WI, USA, # G7570) was used according to the manufacturer’s protocol. Protein levels were determined, and fluorescence was normalized to the amount of protein and to the fed condition.

### 2.11. Statistical Analysis

All data were analyzed using either IBM SPSS Statistics 24 software (IMB Corporation, New York, NY, USA) or GraphPad Prism 8.00 (GraphPad Software, La Jolla, CA, USA). To compare two groups, samples were compared by a non-parametric independent test (Mann–Whitney U-test) as none of the samples followed normal distribution. To compare more than two groups, one-way analysis of variance (ANOVA) was used followed by least significant difference (LSD) post hoc analysis. Unless stated otherwise, all the values are expressed as mean ± standard error of the mean (SEM), *n* = corresponds to number of independent experiments or number of individual measures, and * *p* ≤ 0.05, ** *p* ≤ 0.01, *** *p* ≤ 0.01 and **** *p* ≤ 0.0001 were considered to be significant.

## 3. Results

### 3.1. MERCS-Related Mfn1, Mfn2 and TOM70 Protein Levels are Decreased in Human Post-Mortem Cortex from FAD APP^Swe^ Patients

Our group and others have previously shown that levels of MERCS-related proteins are increased in cortical tissues derived from sporadic AD (SAD) patients [11,40], and it has been suggested that Aβ can have a role in triggering this increase [3,4,5,8,11]. Therefore, we decided to assess MERCS-associated protein levels in human post-mortem frontal cortex from FAD patients carrying *APP* with the Swedish mutation (*APP*^Swe^). This mutation is known to increase the overall level of Aβ and, in particular, the levels of Aβ40 and Aβ42 [41,42]. Our analysis showed that the MERCS-associated proteins Mfn1, Mfn2 and TOM70 were significantly decreased in FAD patients as compared to controls (Figure 1 and Appendix A). We also observed that the level of the inner mitochondrial membrane protein TIM23 was significantly lower than in controls (Figure 1 and Appendix A). Importantly, not all MERCS and mitochondrial proteins were affected (e.g., IP3R3, Grp75, VDAC1, TOM20, VAPB, PTPIP51), suggesting that changes might be connected to a specific type and functions of MERCS (Figure 1 and Appendix A). In addition to their role in regulating ER–mitochondrial connectivity at MERCS, Mfn1 and Mfn2 also have important roles in mitochondrial dynamics [43]. Moreover, it has previously been shown that Mfn1 and Mfn2 protein levels are altered and that there is an increase in mitochondrial fragmentation in SAD patients and AD cell models [40,44]. Since we observed alterations in Mfn1 and Mfn2 protein levels, we decided to also assess the levels of mitochondrial dynamic proteins Opa1 (fusion-related protein) and Drp1 (fission-related protein) in the same human brain samples. No significant changes between control and FAD samples were observed in either Opa1 or Drp1 protein levels (Figure 1 and Appendix A). Therefore, the decrease in Mfn1, Mfn2 and TOM70 protein levels suggests alterations in MERCS and mitochondrial ultrastructure and/or function. However, due to the delicate nature of the material and sample preparation method, it was not possible to assess MERCS and mitochondria ultrastructure nor function in these human post-mortem samples.

### 3.2. Aβ42 Increases the Number of MERCS in Different AD Mouse Models and Primary Cortical Neurons

Due to (i) the dynamic nature of MERCS [9], (ii) increment of MERCS in different AD models [2,4,11,13], (iii) the progression of neurodegeneration and plaques from CA1 to cortical areas in AD [1] and (iv) the possible role of Aβ in modulating MERCS [5,11], we hypothesized that animals of different ages, with different mutations and disease progression rates (tg *App^Swe/Lon^*, knock-in *App^NL-F^* and *App^NL-G-F^*), could differ in their MERCS ultrastructure.

Hippocampus (CA1): analysis of CA1 showed that the number of MERCS per mitochondria was increased in knock-in *App^NL-F^* and *App^NL-G-F^* animals at 10 months (Figure 2A–D). Moreover, while the number of MERCS was increased in *App^NL-F^* mice at 4, 6.5 and 10 months of age, the same increase was observed only in 4 and 10 months old *App^NL-G-F^* mice (Appendix A). However, no changes in MERCS length were detected in these two models (Appendix A). On the other hand, the transgenic model *App^Swe/Lon^* showed an increased MERCS length at 6.5 months while no differences in MERCS number nor MERCS number per mitochondria (Figure 2A and Appendix A) were observed. Interestingly, mitochondrial profile number was increased in both *App^NL-F^* and *App^NL-G-F^* at all measured ages (except in *App^NL-F^* at 4 months) (Appendix A), and the average perimeter per mitochondria profile was decreased in 6.5 months old for *App^NL-F^* and in 10 months for both *App* knock-in models (Appendix A), suggesting an increase in mitochondria fragmentation in these mouse models. Such differences were not detected in *App^Swe/Lon^* mice, neither in mitochondria profile number nor in mitochondria perimeter (Appendix A).

Cortex: In contrast to CA1, a significant increase in the number of MERCS per number of mitochondria was detected in the *App^Swe/Lon^* model at 6.5 months (Figure 2E–G), while the *App^NL-F^* and *App^NL-G-F^* models showed no differences at any age analyzed (Figure 2E). However, MERCS number was significantly increased in all three models at 6.5 months (Appendix A), while MERCS length was significantly decreased in all models at 10 months (Appendix A) when compared to WT. As observed in CA1, mitochondrial profile number was significantly increased in *App^NL^*^-F^ and *App^NL-G-F^* mice at all ages (Appendix A) and mitochondria perimeter decreased at 10 months (Appendix A).

App^NL-F^-derived PCN: Next, we assessed mitochondria and MERCS ultrastructure in PCN derived from *App^NL-F^* mice. Since it has been shown that both the *App* KI models present increased levels of Aβ42 over Aβ40 [34], we started by confirming if also *App^NL-F^* embryonic neurons produce high levels of Aβ42. As described in adult animals [34], *App^NL-F^* PCN showed an increase in extracellular Aβ42 levels and a decrease in extracellular Aβ40 levels as compared to WT PCN (Appendix A). Like what was observed in vivo (Appendix A), MERCS and mitochondria profile number were increased in *App^NL-F^*-derived PCN as compared to WT PCN (Figure 2H–I,L). However, when the number of MERCS was corrected to mitochondrial profile number, we observed no difference between WT and *App^NL-F^* derived PCN (Figure 2J,L). Still, the percentage (%) of mitochondria surface in contact with ER was increased in *App^NL-^*^F^-derived cells (Figure 2K,L), indicating that longer contacts are formed between the two organelles in these neurons. In the past years, the role of Mfn2 in regulating MERCS ultrastructure has been widely discussed. While some studies report that Mfn2 is important for the tethering of the two organelles [45,46], we and others have shown that Mfn2 acts as a negative regulator of these contacts [5,17]. Since Mfn2 levels have been reported to be altered in SAD [40] and FAD (Figure 1) patients, we decided to investigate whether the levels of these proteins were altered in *App^NL-F^*-derived PCN. Indeed, levels of Mfn2 were significantly decreased, while levels of VDAC1 were not altered (Figure 2M and Appendix A).

WT-derived PCN treated with synthetic Aβ42: For several years, it was thought that amyloid plaques are responsible for the neurodegenerative process in AD, however, recent evidences show that oligomeric Aβ (oAβ) correlates better with the cognitive decline observed in patients [47]. Recently, we have shown that the presence of extracellular amyloid plaques in human brain biopsies do not affect MERCS number or length [8]. Therefore, we decided to study if Aβ42 itself as well as its different aggregation forms (monomeric Aβ42 (mAβ42) and oAβ42) had an effect in MERCS by incubating WT PCN with synthetic Aβ42 and assessed MERCS and mitochondria ultrastructure (Appendix A). Firstly, we performed a dose-response curve to determine if and at which concentration Aβ42 affects MERCS (Appendix A). We decided to use Mfn2 as a read-out due to its role in MERCS ultrastructure modulation. Since incubation of WT PCN with 2 µM Aβ42 lead to changes in Mfn2 levels, indicating a potential alteration in MERCS (Appendix A), we decided to use this concentration for our analysis. Our data showed that different aggregation forms of Aβ42 affect MERCS differently (Figure 2N,O). While mAβ42 increased MERCS number, MERCS number per mitochondria profile and % of mitochondria surface in contact with ER (Figure 2N,O and Appendix A), oAβ42 only increased the number of MERCS per mitochondrial profile (Figure 2O and Appendix A), most likely due to a slight non-significant reduction in mitochondrial profile number (Appendix A). In contrast to previous data, no significant differences were found in mitochondrial number or perimeter (Appendix A). scFvA13 is an antibody fragment that recognizes and neutralizes oAβ42 and, if expressed as intrabody, is also able to selectively neutralize oAβ42 and functionally silence it [38,39,48]. When a mixture of mAβ42 and scFvA13 was applied to neurons, no differences in MERCS number, MERCS number per mitochondria profile and % of mitochondria surface in contact with the ER were detected as compared to mAβ42 alone (Figure 2N,O and Appendix A). However, when a mixture of oAβ42 and scFvA13 was applied to the neurons, the number of MERCS per mitochondria were the same as in control, confirming both the identity of the aggregation species and the specific effect of oAβ42 in MERCS (Figure 2O and Appendix A). In addition, WT PCN treated with mAβ42 showed a significant decrease in Mfn2 levels, while VDAC1 was not altered (Figure 2P and Appendix A). Altogether, these data suggest that Aβ42 affects MERCS and mitochondria ultrastructure and that these changes can be connected to Mfn2 levels or to the function of the protein. These data also suggest that altered mitochondria ultrastructure and increased MERCS are early events detected at embryonic stages and directly triggered by Aβ42.

### 3.3. Early Activation of Autophagosome Formation in Starved Primary Cortical Neurons Derived from App^NL-F^ Mice

As mentioned before, we know that autophagosomes can be formed at MERCS [23,24], modulation of MERCS affect autophagosome formation [23,26], MERCS are altered in *App^NL-F^*-derived PCN (Figure 2) and FAD as well as SAD patients show an increase in autophagic vesicles and a defect in the autophagy pathway [20,49]. This accumulation of AVs can be due to either an increase in autophagosome formation or impaired maturation of autophagosomes. Since we knew that MERCS were altered in *App^NL-F^*-derived PCN, we decided to assess autophagosome formation using starvation, which is a well-established method to induce autophagy [35]. Autophagosome formation was assessed as levels of microtubules-associated protein 1A/1B-light chain 3B (LC3B) and SQSTM1/p62 (p62) protein levels during starvation. LC3B is synthetized in an immature form which, after being cleaved, forms LC3B-I, that can be conjugated with phosphatidylethanolamine to form LC3B-II. LC3B-II is one of the few proteins that remain attached to the autophagosome upon its maturation, only being degraded upon the formation of autolysosome (organelle resulting from the fusion of autophagosome and lysosome). Therefore, the levels of LC3B-II as well as the ratio between LC3B-II/I have been widely used as markers for autophagosomes and autophagy flux, respectively [35]. p62 levels are an indirect measure of autophagic activity since it mediates the binding of ubiquitinated cargo targeted for degradation by autophagy to LC3, allowing their engulfment by the organelle. Under normal conditions, upon induction of autophagy, a decrease in p62 levels is observed, since p62 itself is degraded in the process.

In WT PCN, we observed an increase in LC3B-I, LC3B-II and LC3B II/I at 1.5 and 2 h of starvation, followed by a decrease in these protein levels (Figure 3 and Appendix A). However, for *App^NL-F^*-derived PCN, this increase occurred at 1 and 1.5 h (Figure 3 and Appendix A). Furthermore, there is a significant decrease in p62 in starved WT PCN (Figure 3 and Appendix A), while no differences in p62 were found in starved *App^NL-F^* PCN (Figure 3 and Appendix A). The unchanged p62 levels in *App^NL-F^* neurons upon starvation could be due to either p62-cargo mistargeting, or impaired end stage autophagosmal/lysosomal and p62 degradation (Figure 3 and Appendix A). Bafilomycin A1 (Baf) is an inhibitor of the fusion between autophagosomes and lysosomes, which leads to the accumulation of LC3-II due to the inability of immature autophagosome to degrade its cargo by itself [35]. Upon Baf treatment, we observed an increase in LC3B-II at the different starvation time points, when compared to non-treated cells at the same time point, in both WT and *App^NL-F^* PCN (Appendix A). Therefore, our data suggest that the fusion between autophagosome and lysosome is functional in our starved PCN.

### 3.4. WT- and App^NL-F^-Derived PCN Show That by Increased Mitochondrial ER, Connectivity Precedes Autophagosome Formation

Even though it has been demonstrated that isolation membranes, which later mature and form autophagosomes, can have their origin from MERCS [23,24], it remains to elucidate whether MERCS ultrastructure is altered during autophagosome formation induced by starvation. We observed that the number of MERCS were significantly increased after 1.5 h of starvation and significantly decreased at 2 h in WT PCN. However, even though a similar trend was observed in *App^NL-F^*-derived PCN, this pattern was shifted half-hour/one hour earlier (0.5 h/1 h) when compared to WT PCN. After 1.5 h of starvation, the number of MERCS decreased back to fed levels (Figure 4A) in *App^NL-F^*-derived PCN. Interestingly, the number of mitochondrial profiles was also increased at the same time as MERCS were increased, i.e., before the increase of the LC3II/I ratio in both models (Figure 4B). Due to mitochondrial profile number variation, we decided to normalize the number of MERCS to the number of mitochondrial profiles (Appendix A). We observed that in both WT and *App^NL-F^* neurons, the number of MERCS per mitochondria was increased after 1.5 and 0.5 h of starvation respectively (Figure 4A), right before the significant increase in LC3B-II/I (Figure 3). We also observed an increase in MERCS length and % of mitochondria surface in contact with ER at 1 h in *App^NL-F^* PCN and 1.5 and 2.5 h in WT PCN (Appendix A), while mitochondria perimeter was decreased during the first 1 h after starvation in both cell models (Appendix A). Due to the important role of Mfn2 in regulating MERCS ultrastructure, we decided to investigate the levels of this protein during starvation. Indeed, in both WT- and *App^NL-F^*-derived PCN, we observed a significant increase in Mfn2 levels at the same time as the reduction of MERCS number, and mitochondria fragmentation was observed (Figure 4C).

### 3.5. Increased Mitochondrial Function and ATP Levels Precedes Autophagosome Formation During Starvation in Both WT and App^NL-F^ PCN

Previously, it has been reported that alterations in connectivity between mitochondria and ER can affect mitochondrial ATP formation since the activity of several dehydrogenases involved in tricarboxylic acid cycle are regulated by Ca^2+^ [32,50,51,52,53,54,55]. Since our data show alterations in mitochondria ultrastructure and Mfn2 protein levels, we decided to investigate mitochondrial activity/function by measuring oxygen consumption rate (OCR), mitochondrial membrane potential and total ATP levels. We started by investigating if *App^NL-F^* or WT neurons treated with Aβ42 had alterations in OCR as compared to non-treated WT neurons. In the different models and conditions, we observed an increase in basal respiration, ATP production and maximal respiration (Appendix A), where connectivity between ER and mitochondria were previously shown to be increased (Figure 2H,K–L,N–O), showing that these parameters are already altered before starvation. This increase in mitochondrial function is likely related with increased mitochondria–ER connectivity and can be interpreted as an initial compensatory response to Aβ accumulation. In fact, increased expression of mitochondrial genes have been reported in patients with mild cognitive impairment, a pre-stage of AD [56]. Then, we assessed the same parameters in starved cells and observed a significant increase in basal respiration from 1 h (Figure 4D, middle left panel) and an increase in ATP production at 1 and 1.5 h in WT PCN (Figure 4D, middle right panel), as well as a decrease in maximal respiration at all time points (Figure 4D, right panel). In *App^NL-F^*-derived PCN, similarly to the ultrastructural data, we detected an increase in basal respiration at 0.5 h and ATP production at 0.5 and 1 h, while maximal respiration had a tendency to be increased at 0.5 h and it was significantly decreased at 2.5 h (Figure 4E). Since changes in mitochondrial respiration can lead to changes in mitochondrial membrane potential, we decided to measure the amount of TMRM released from mitochondria upon depolarization with FCCP in the presence of complex V inhibitor oligomycin. TMRM is a positively changed fluorescent probe and its retention by mitochondria is used to estimate changes in mitochondrial membrane potential. In line with the OCR data, *App^NL-F^* PCN displayed an earlier increase in mitochondrial membrane potential (0.5 h) as compared to WT PCN (0.5 to 2 h) (Figure 4F,G). Since mitochondrial respiration and membrane potential were increased, we decided to investigate if total ATP levels were altered during starvation. Indeed, total ATP levels were increased at 1 h in WT-derived PCN while, as before, *App^NL-F^* PCN showed an increase in total ATP levels at 0.5 h (Figure 4H).

In summary, we report that the proximity between the mitochondria and ER increases upon increased levels of Aβ or starvation. During starvation, the increased formation of MERCS leads to alterations in mitochondrial function and increased total ATP production, which has been shown to be essential for autophagosome formation. Moreover, we show that under starvation, this cascade of events is induced at earlier time-points in *App^NL-F^* neurons as compared to WT neurons, indicating that the presence of Aβ induces a quicker response.

## 4. Discussion

In the past few years, evidence supporting the MERCS hypothesis of AD has been emerging [57,58]. Different AD mouse and cell models have shown upregulation of MERCS-related functions as well as proximity between ER and mitochondria, suggesting a dysregulation of this subcellular region in AD [11,59,60]. However, most of these previous studies relied on non-neuronal models (including patient-derived cells) or neuronal models overexpressing APP and its fragments [4,13,59,61,62]. Therefore, the exact causes and consequences of alterations in MERCS ultrastructure in AD neuronal tissue remain elusive. Here, we report that Aβ per se led to the increment of the juxtaposition between ER and mitochondria, in turn affecting mitochondrial function and autophagosome formation in PCN.

Importantly, we aimed to investigate variations in MERCS ultrastructure during disease progression and therefore, we analyzed two brain regions (CA1 and cortex) obtained from different AD animal models of several ages. The three AD animal models used recapitulate amyloid pathology either through APP overexpression (*App^SweLon^*) or knock-in (*App^NL-F^* and *App^NL-G-F^*) of FAD APP mutations known to increase the Aβ42/Aβ40 ratio. In these AD animal models, we observed an increased number of MERCS per cell in CA1 before plaque formation (*App^NL-F^*) or memory impairment (*App^NL-G-F^*), suggesting that the contact homeostasis is altered before these hallmarks arise (Appendix A). However, when we accounted for the number of contacts for mitochondria fragmentation (observed at 10 months by increase in mitochondria profile number and decrease in mitochondria profile perimeter (Appendix A)), we only observed an increase in numbers of MERCS per mitochondria at 10 months, suggesting that this phenotype comes later (Figure 2A). Therefore, our data suggest that MERCS alteration occur prior to alterations on mitochondria dynamics. In fact, it has been proposed that ER wraps around the exact site where mitochondria undergoes fission, forming a contact site, by recruiting calcium-sensitive ER-bound protein inverted formin 2 and mitochondrial protein actin-nucleating [63,64,65], showing that this increase in MERCS observed at early stages in the models analyzed could influence mitochondria dynamics. In addition, the OE *App^Swe/Lon^* model only showed alterations in MERCS ultrastructure in cortex, while the *App* KI models showed most of the alterations in the CA1 (Figure 2A–G). The described differences in MERCS number and mitochondrial fragmentation as well as regional differences can be explained by the fact that the *App^Swe/Lon^* mouse model overexpresses full-length APP and thus generates increased levels of APP fragments, while *App^NL-F^* or *App^NL-G-F^* mice express physiological levels of APP. In fact, the accumulation of the transmembrane fragment C99 (which originate after the first cleavage of APP by β-secretase) has been shown to increase the connectivity between ER and mitochondria and to affect cholesterol, lipid droplets and sphingolipid metabolism, showing that the increase in one of the APP fragments can affect the contact per se [66]. Therefore, wary conclusions should be drawn from results obtained from animal models overexpressing APP since we are unaware how APP cleavage fragments can affect cell metabolism and MERCS ultrastructure. Surprisingly, both the *App* KI models presented similar MERCS and mitochondria phenotypes even though the arctic mutation in *App^NL-G-F^* increases the amounts of plaque formed in these animals (Figure 2A–G and Appendix A), again supporting MERCS alteration as an early event prior to plaque formation. For several years, amyloid plaques were believed to be one of the major causes of neuronal death but, more recently, levels of oligomeric species of Aβ have been described to better correlate with neuronal death and disease progression [67]. Indeed, we have previously shown that the presence of amyloid plaques alone does not alter MERCS ultrastructure in human biopsy samples [8]. Here, we report increased mitochondria–ER apposition in PCN derived from *App^NL-F^* mice and WT PCN incubated with mAβ42 and oAβ42 (Figure 2H,K,N,O). We further prove that the alterations in MERCS are Aβ-specific since cells co-treated with oAβ42 and the oAβ-neutralizing antibody fragment scFvA13 displayed similar MERCS ultrastructure as the controls (Figure 2N–O and Appendix A). It should be noted here that the aggregation forms of Aβ42 were not assessed after the 24 h incubation with PCN and, therefore, we cannot exclude that oAβ42 were formed from mAβ42, either in the cell media or intracellularly during this incubation time. However, it is still unknown if it is intra- or extra-cellular Aβ that affects the connectivity between ER and mitochondria. Previously, it has been shown that both the Swedish and London mutation in APP increase the concentration of its extracellular soluble fragment, while intracellular fragments were not altered, suggesting that extracellular Aβ could affect MERCS ultrastructure [68]. In fact, PCN derived from *App^NL-F^* secrete more Aβ42 as compared to WT PCN (Appendix A). Thus, the increase in MERCS in both PCN derived from *App^NL-^*^F^ and WT PCN treated with mAβ42 and oAβ42 suggest that extracellular Aβ can affect MERCS. However, further experiments are needed to unravel the detailed mechanism as well as the specific roles of intra- and extra-cellular Aβ.

In parallel to these experiments, we have also assessed the levels of several mitochondrial and MERCS-associated proteins in human *APP^Swe^* post-mortem brain tissue and detected significantly decreased levels of Mfn1 and Mfn2 (Figure 1). Like these results, also the levels of Mfn2 were decreased in *App^NL-F^* PCN and in WT PCN exposed to mAβ42. Since Mfn2 has been described as a negative regulator of MERCS [5,17,18], in addition to its well-established role in mitochondrial fusion [43], our data suggests that the increased connectivity between ER and mitochondria in these AD models due to the increase in Aβ could be linked with reduced Mfn2 levels. In fact, MERCS has been previously shown to be increased in models with FAD mutation in Presenilin 2 (PS2), an essential component of γ-secretase, due to the tuning of the antagonistic effect of Mfn2 via direct interaction between these two proteins [12]. Moreover, the role of Mfn2 in neurons appears to be of fundamental importance in cell physiology and in neuropathologies [69,70,71]. Indeed, a recent study showed that conditional KO of Mfn2 in adult mouse forebrain neurons led to alterations of mitochondria dynamics and distribution as well as increased apoptosis in hippocampal and cortical neurons. Moreover, the authors also showed loss of neuronal marker NeuN in CA1 in concomitance with increase in oxidative damage and cortical atrophy [72]. Interestingly, this phenotype is very similar to what has been described in AD and AD models [1,73]. Furthermore, Calvo-Rodriguez and colleagues showed that rat primary hippocampal neurons incubated with oAβ42 presented an increased connectivity between ER and mitochondria, alterations in Ca^2+^ release by ER and uptake by mitochondria and increased cell death [74]. This could explain why we observed the increase in mitochondrial function in WT PCN treated with synthetic Aβ42 (Appendix A). Since the fine-tuning of Ca^2+^ within mitochondria is essential for cellular homeostasis (low concentration of Ca^2+^ boosts mitochondria function but, in excess, triggers apoptosis) [57,58], we hypothesize that Aβ42 increases MERCS ultrastructure/function, either directly or indirectly, by affecting MERCS-associated proteins such as Mfn2, which ultimately could be connected to cell death by the excess of Ca^2+^ transferred into mitochondria followed by induction of apoptosis.

We then studied the interplay between Aβ, MERCS and autophagy. Autophagosome maturation has been shown to be impaired in SAD and FAD post-mortem brain [20,49] with accumulating AVs containing Aβ aggregates. Previously, it was shown that knock-down of Mfn2 and overexpression of VAPB and PTPIP51 leads to an increase in MERCS and reduction of the pool of autophagosomes [23,24,26]. In concordance, we observe a sequential event during starvation of PCN, starting with increased MERCS, followed by induction of autophagy and increased Mfn2 levels, leading to a negative regulation of MERCS (Figure 3 and Figure 4). Importantly, these sequential events occur earlier in the *App^NL-F^* model as compared to WT (Figure 5). Therefore, we postulate that Mfn2 can negatively regulate autophagosome formation, probably in connection with alterations in MERCS ultrastructure and/or functions. Studies supporting both cases have been published. For example, modulation of Ca^2+^ release from the ER affects autophagosome formation, e.g., modulating the AMPK signaling pathway via ATP production [27,28,29,30,75,76]. In fact, during the first phases of starvation, we observed increased mitochondrial function (basal respiration, ATP production, membrane potential) and cellular ATP levels (Figure 4D–H), in line with studies reporting that acute amino acid starvation can result in increased ATP levels and mitochondrial function [30,31]. Although an increase in ATP is usually associated with inhibition of autophagy due to dephosphorylation of AMPK, it has been shown that it can also have a direct role in the formation of autophagosomes [30]. The initial increase in mitochondrial function was subsequently abolished upon formation of autophagosomes, which we believe is related to a rescuing mechanism to prevent excessive autophagosome formation regulated by Mfn2 and MERCS, although further experimental data is needed to prove these data. In fact, our hypothesis that Mfn2 regulates autophagy via MERCS is further corroborated by a recent paper showing that increase in energy stress (by starvation or blocking complex I) results in the translocation of AMPK into MERCS, allowing it to interact and phosphorylate Mfn2, culminating in increased mitochondria fission and autophagy [77]. Our data is further supported by the fact that autophagosome half-life is about 20–30 min in mammalian cells [78], which could explain why MERCS are increased 30 min before the increase in LC3-II (Figure 3). Thus, we believe that the increase in MERCS observed in *App^NL-F^* PCN leads to a faster increase in mitochondrial function and ATP, allowing for faster formation of autophagosomes as compared to WT PCN. In the future, it would be interesting to explore if this mechanism could be connected to a cellular feedback response to increased toxic Aβ levels, leading to an attempt to increase the neuronal energy level and self-cleaning capacity to avoid excessive Aβ aggregation.

## 5. Conclusions

In conclusion, we show that Aβ induces the juxtaposition of ER and mitochondria, which in turn affects the formation of autophagosomes and mitochondrial function during starvation. However, the exact molecular mechanisms of how ATP and MERCS induce or prevent autophagosome formation, maturation and fusion with lysosomes needs further exploration both in health and disease. Our data helps to better understand how two different biological processes are connected in models with increased Aβ. This data suggests modulation of MERCS as a possible drug target, helping to regulate multiple biological processes affected in AD.

## Figures and Tables

**Figure 1 cells-09-02552-f001:**
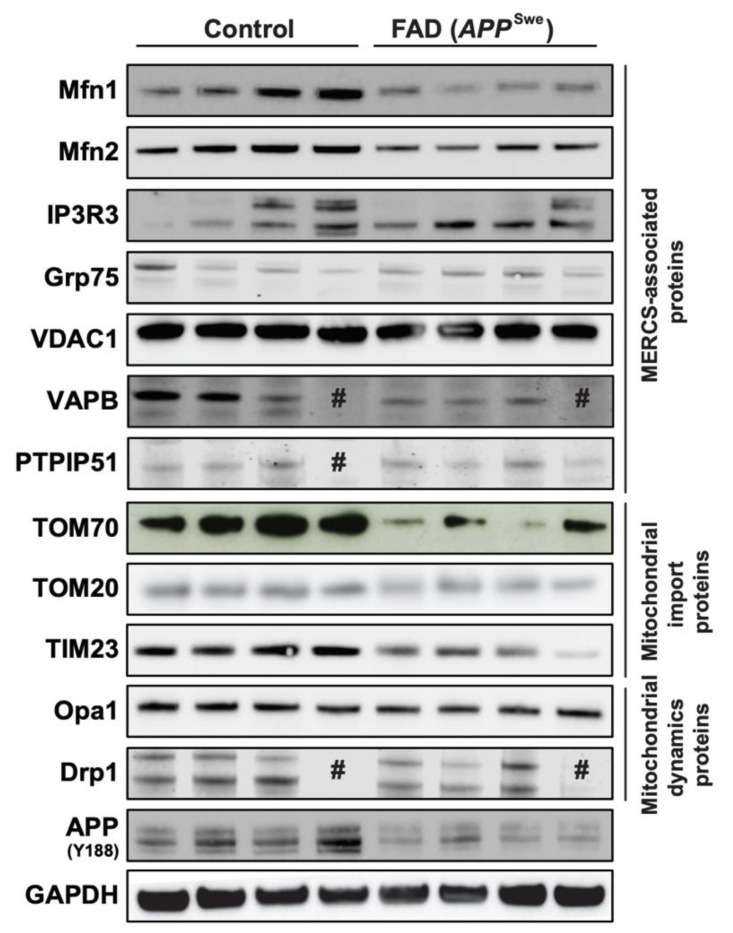
Mitochondria-endoplasmic reticulum (ER) contact sites (MERCS)-associated proteins Mitofusin 1 (Mfn1), Mitofusin 2 (Mfn2) and translocase of the outer mitochondrial membrane 70 (TOM70) are altered in familial Alzheimer’s disease (FAD) post-mortem brain. Post-mortem frontal cortices were homogenized, sodium dodecyl sulphate–polyacrylamide gel electrophoresis was performed, and proteins were stained as indicated. Representative immunoblots of MERCS-associated proteins, mitochondrial import proteins and mitochondrial dynamics proteins. # sample loaded but no band observed (*n* = 3–4).

**Figure 2 cells-09-02552-f002:**
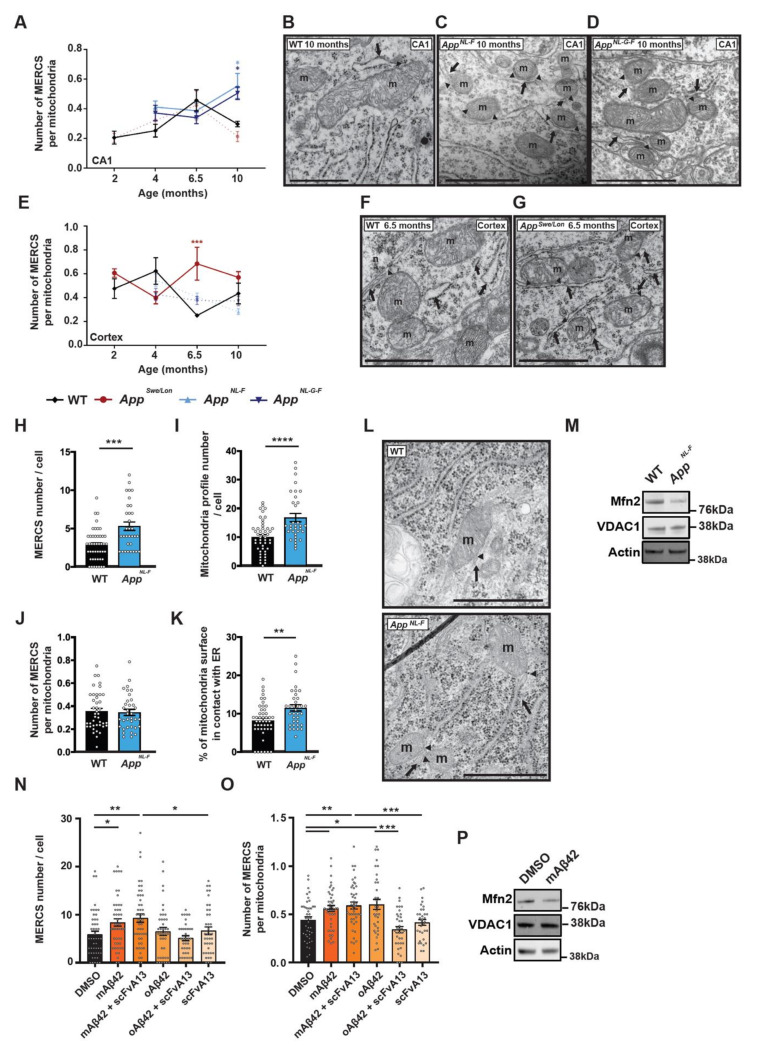
MERCS are increased in Alzheimer’s disease (AD) mouse models and primary cortical neurons (PCN) with increased amyloid β-peptide (Aβ42). Quantification of number of MERCS per mitochondria in (**A**) CA1 (*p* = 0.021 *App^NL-F^*, *p* = 0.051 *App^NL-G-F^*)—electron micrographs from 10 months animals, (**B**) wild-type (WT), (**C**) *App^NL-F^*, (**D**) *App^NL-G-F^* and (**E**) cortex (*p* = 0.001)—electron micrographs from 6.5 months animals. (**F**) WT and (**G**) *App^Swe/Lon^*. *App^Swe/Lon^* (red circles), *App^NL-F^* (light blue up-triangle) and *App^NL-G-F^* (dark blue inverted-triangle) and respective WT control (black rhombus). Solid and dotted lines were used for better visualization when data was non-significant but represent the same animals in both (**A**) and (**E**). Values represent average of *n* = 3 (WT and *App^Swe/Lon^*) or *n* = 4 (*App^NL-F^* and *App^NL-G-F^*) animals and each animal model was compared to age-matched WT. Each animal value was obtained by randomly selecting 3 pictures out of >100 pictures per animal and all mitochondria and MERCS quantified. Quantification of (**H**) MERCS number (*p* = 0.0003), (**I**) mitochondria profile number (*p* = 0.0001), (**J**) number of MERCS per mitochondria, (**K**) % of mitochondria surface in contact with ER (*p* = 0.0064) and (**L**) respective electron micrographs from WT- or *App^NL-F^*-derived PCN. Each dot represents a measurement of a single cell. 33 ≤ *n* ≤ 47 from 8 (WT) or 5 (*App^NL-F^*) independent experiments. (**M**) Representative immunoblot for labelled proteins. Quantification of (**N**) MERCS number (*p* = 0.0117 DMSO vs. mAβ42; *p* = 0.0025 DMSO vs. mAβ42 + scFvA13; *p* = 0.0367 mAβ42 + scFvA13 vs. scFvA13), and (**O**) MERCS number per mitochondria (*p* = 0.0215 DMSO vs. mAβ42; *p* = 0.0052 DMSO vs. mAβ42 + scFvA13; *p* = 0.0006 mAβ42 + scFvA13 vs. scFvA13; *p* = 0.0335 DMSO vs. oAβ42; *p* = 0.0003 oAβ42 vs. oAβ42 + scFvA13) from electron micrographs from PCN incubated with 2 µM of monomeric Aβ42 (mAβ42), mAβ42 and anti-oAβ single-chain antibody fragment A13 (scFvA13) (mAβ42 + scFvA13), oligomeric Aβ42 (oAβ42), oAβ42 with scFvA13 (oAβ42 + scFvA13) and with only scFvA13. Each dot represents the measurement of a single cell (33 ≤ *n* ≤ 49). Total of 5 independent experiments. (**P**) Representative immunoblot for labelled proteins of WT PCN. Scale bars corresponds to 500 nm, m—mitochondria, arrow—ER, arrow heads—MERCS, *n*—nucleus. *P*-values were obtained by using one-way ANOVA and LSD post hoc analysis for (A) and (E) and non-parametric independent Mann–Whitney U test (comparison to WT or DMSO) in (H–K) and (N, O). * *p* ≤ 0.05, ** *p* ≤ 0.01, ***, *p* ≤ 0.01 and **** *p* ≤ 0.0001 were considered significant.

**Figure 3 cells-09-02552-f003:**
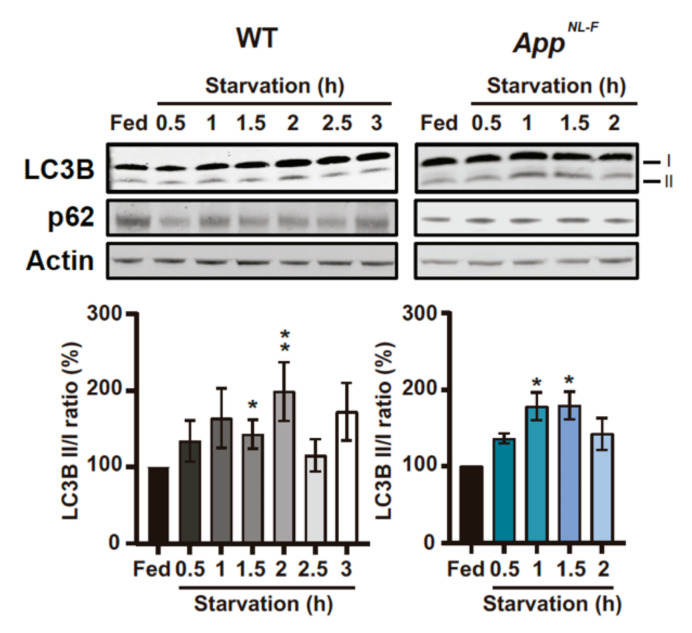
Levels of microtubules-associated protein 1A/1B-light chain 3B (LC3) and SQSTM1/p62 (p62) are altered in WT and *App^NL-F^* PCNduring starvation. Representative immunoblots of LC3B and p62 in starved 14 days *in vitro* (DIV) PCN derived from WT and *App^NL-F^* and respective quantification of LC3B II/I ratio (WT: *p* = 0.0272 Fed vs. 1.5 and *p* = 0.0013 (3 ≤ *n* ≤ 10); *App^NL-F^ p* = 0.0286 Fed vs. 1 and Fed vs. 1.5 (*n* = 4)). *P*-values were obtained by using non-parametric independent Mann–Whitney U test (comparison to Fed). * *p* ≤ 0.05 and ** *p* ≤ 0.01 were considered significant.

**Figure 4 cells-09-02552-f004:**
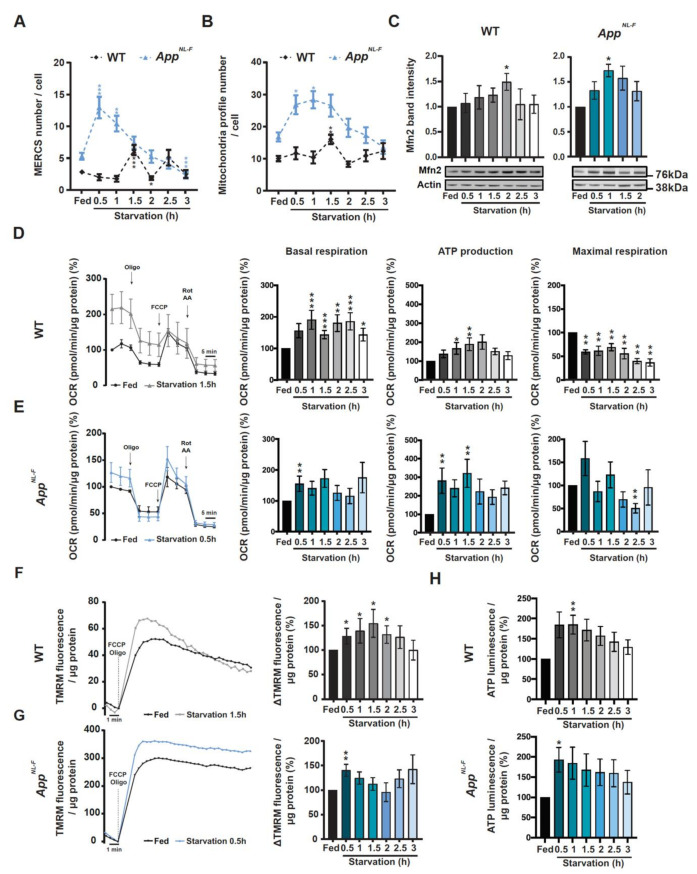
MERCS and mitochondria ultrastructure as well mitochondrial respiration, membrane potential and adenosine triphosphate (ATP) formation are altered during starvation in a time-dependent manner. Quantification of (**A**) MERCS number (WT: *p* = 0.0004 Fed vs. 1.5, *p* = 0.0141 Fed vs. 2; *App^NL-F^*: *p* = 0.001 Fed vs. 0.5, *p* = 0.0024 Fed vs. 1, *p* = 0.0008 Fed vs. 3) and (**B**) mitochondrial profile number (WT: *p* = 0.0012; *App^NL-F^*: *p* = 0.0183 Fed vs. 0.5, *p* = 0.0012 Fed vs. 1) (12 ≤ *n* ≤ 47 from 8 (WT) or 5 (*App^NL-F^*) independent experiments). (**C**) Representative immunoblots and respective quantification of Mfn2 during starvation in WT (left panel) and *App^NL-F^* (right panel) (WT: *p* = 0.0013 Fed vs.; *App^NL-F^*: *p* = 0.0286 Fed vs. 1 and Fed vs. 1.5) (3 ≤ n ≤ 6 (WT) and 4 (*App^NL-F^*) independent experiments). (**D**) Representative oxygen consumption rate (OCR) trace of starved WT PCN and respective measurements of basal respiration (*p* = 0.0002 Fed vs. 1, Fed vs. 1.5 and Fed vs. 2.5; *p* = 0.0057 Fed vs. 2; *p* = 0.0101 Fed vs. 3), ATP production (*p* = 0.101 Fed vs. 1, *p* = 0.0057 Fed vs. 1.5) and maximal respiration (*p* = 0.0022) (*n* = 7–8). (**E**) Representative oxygen consumption rate plot of starved *App^NL-F^* PCN and respective measurements of basal respiration (*p* = 0.0079), ATP production (*p* = 0.0079) and maximal respiration (*p* = 0.0079) (*n* = 5). Tetramethylrhodamine, methyl ester (TMRM) fluorescence trace after induction of its release by carbonyl cyanide-4-(trifluoromethoxy) phenylhydrazone (FCCP) (2.5 µM) and Oligomycin (3.2 µM) of starved (**F**) WT PCN (*p* = 0.0169, *n* = 7) and (**G**) *App^NL-F^* PCN (*p* = 0.0079, *n* = 5). (**H**) Total ATP levels of starved WT (*p* = 0.0079, *n* = 5) and *App^NL-F^* PCN (*p* = 0.0286, *n* = 4). *P*-values were obtained by using non-parametric independent Mann–Whitney U test (comparison to respective Fed condition). * *p* ≤ 0.05, ** *p* ≤ 0.01, *** and *p* ≤ 0.01 were considered significant.

**Figure 5 cells-09-02552-f005:**
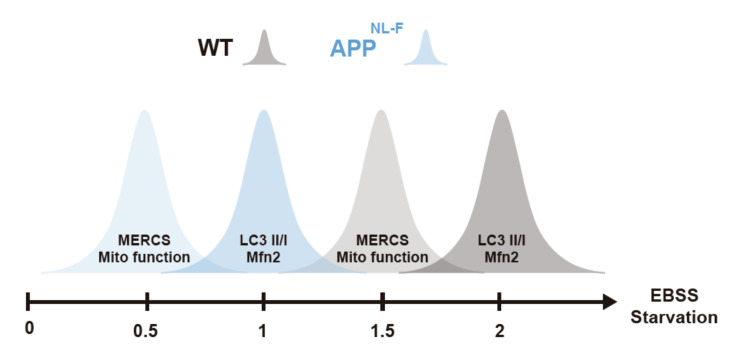
Increase in connectivity between endoplasmic reticulum (ER) and mitochondria as well as mitochondria function occurs before autophagosome formation. Graphical representation of alterations of MERCS and mitochondria ultrastructure/function, autophagosome formation and Mfn-2 levels during starvation. Starvation axis represents starvation times in hours.

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
