# Peer review of "Amyloid β-Peptide Increases Mitochondria-Endoplasmic Reticulum Contact Altering Mitochondrial Function and Autophagosome Formation in Alzheimer’s Disease-Related Models"

_cells, 2020, doi:10.3390/cells9122552_

Round 1

Reviewer 1 Report

The article „Amyloid β-peptide increases mitochondria-endoplasmic reticulum contact altering mitochondrial function and autophagosome formation in Alzheimer’s disease related models“ (Leal et al.) describes how Ab-derived pathophysiology affects mitochondria-ER contact sites (MERCS) related to mitochondrial function and autophagy. For these experiments they used human post mortem tissue from FAD APPswe patients and primary cultured neurons from different AD mouse models. They found an increased connectivity between ER and mitochondria in neurons of AD mouse models as well as in WT neurons exposed to Ab, whereby that connectivity occurs simultaneously with an increase in mitochondrial activity and is followed by upregulation of autophagosome formation. The authors suggest that Ab and Ab-derived pathophysiology modulates MERCS and thereby mitochondrial activity and autophagy.

There is ample evidence that mitochondria are highly involved in mediating neurodegenerative processes during AD. Since it is still a matter of debate how Aβ interacts with the power supply the presented experiments seem reasonable. The MS is well written and the results in the present study provide new aspects about the molecular details regarding mitochondrial activity and hence also autophagy modulated by Ab.

However, some questions remain e.g. regarding the selected Ab concentration:

  1. The authors found that in post-mortem familial AD (FAD) patients’ brain present alterations in the MERCS-related proteins Mfn1/2 and TOM70. However, these results were not described in the abstract.
  2. Cultured cells from AD mouse models were incubated with different Ab forms at concentrations (1µM) far beyond clinical levels. What is the idea for this unphysiological amounts? Did the authors once try using concentrations in the lower nM range?
  3. To WT cell culture 2µM synthetic Ab has been applied. Why these different concentrations? For the reader, it was slightly confusing.
  4. In the results section the authors described that they performed previous to their experiments with WT cultured neurons a dose response curve with Ab and decided to use 2µM according to its effect on Mfn2. However, as discussed above, 2µM is a pretty high concentration putting the validity of the data severely into question.
  5. MERCS number is clear, but what exactly is a MERCS length? The length ER and mitochondria are in contact?
  6. The experiments with the combined application of scFvA13 and mAb imply that after the (unfortunately not indicated) incubation time, no oligomers have been formed. Actually, according to the chemical properties of Ab, those molecules should oligomerise.
  7. The starvation experiments are not clear to me. How are they set into relation to AD?
  8. In the discussion section, the authors stated that they “observed an increased number of MERCS per 483 cells in CA1 before plaque formation (AppNL-F) or memory impairment (AppNL-G-F), suggesting that the 484 contact homeostasis is altered before these hallmarks arise (Fig. S2A and E)”, however, they performed only cell culture experiments and cannot set behaviour simply in relation to those results.

Author Response

1.We appreciate the comment from the reviewer and we have now added these missing results to the abstract in lines 27-28 – “We start by showing that the levels of Mitofusin 1, Mitofusin 2 and mitochondrial import receptor subunit TOM70 are decreased in post-mortem brain tissue derived from familial AD.”

2-4. The reviewer’s comment is fair, and we apologize for the mistake and confusion caused by the discrepancy between the concentration of Aβ used. In line 303, where it was written 1µM, should have been 2µM, since this is the concentration used throughout all the Aβ experiments. By changing this in the text it should avoid the confusion the reviewer is mentioning above.

Regarding Figure 2 and S3 we would like to point out that this was an acute treatment and the aim was to assess if Aβ by itself, in different aggregation forms, had an effect in MERCS ultrastructure and not necessarily mimic the disease conditions. In fact, we did a dose-response curve and found that exposure with 2µM Aβ resulted in reduced levels of Mfn2 accompanied by an effect on MERCS (Fig S3B and Fig 2 H-L). In addition, the effects observed with and without the intrabody scFvA13, allows us to conclude that oAβ per se can increase the connectivity between ER and mitochondria. We have now altered the text in the result section to make this clearer (line 340-341).

5. We appreciate the input and have now added extra information in Material and Methods explaining how MERCS length was measured (line 178-181).

6.By lapse, the incubation time with the synthetic Aβ was missing in the Material and Methods. We thank the reviewer for noticing this. We have now added this incubation time in line 164.mAβ and oAβ were prepared using two different protocols as described before (Fa et al., 2010; Blaine Stine Jr et al., 2003). When we state mAβ and oAβ we are referring to the species existing in the solution applied to the neurons and we did not follow the aggregation process during the 24h incubation in cell media. This is now better explained in the Material and Methods section (line 156-157). At the same time, we are not claiming that scFvA13 prevents mAβ oligomerisation. However, we would expect that any extracellular oAβ formed during the 24h incubation time in cell media should be functionally neutralised by the intrabody (ref Meli et al., Nature Comms 2014), while the mAb can still affect MERCS. This is partly proven by the fact that MERCS are still up regulated in WT primary neurons treated with mAb, even upon treatment with scFvA13, but not with both oAβ and scFvA13. This is now discussed on lines 524-527.

7. We thank the reviewer for this comment. We have explained why starvation was used to induce autophagy in the beginning of section “3.3. Early activation of autophagosome formation in starved primary cortical neurons derived from AppNL-F mice” but it seems like it was not clear. Autophagosomes has been shown to accumulate in AD. However, the reason of this accumulation is still unknown. Two hypotheses have been purposed: overproduction of “new” autophagosome or impaired maturation and, therefore, accumulation of non-mature autophagosomes. Moreover, we detected an increase of MERCS in our AD model, which in turn has been shown to regulate autophagosome formation. Therefore, to assess autophagosome formation in the AppNL-F PCN, we used starvation as a tool to induce autophagosome formation. We have now changed the text accordingly to make it clearer for the reader - lines 148-150, 367-373).

8. We apologize if this was not clear. This data is referring to the MERCS ultrastructural analysis in AD animal cortex and CA1, not primary culture. We have now changed this sentence, adding this information explicitly (line 491).

Reviewer 2 Report

In this interesting study, authors use several models including brain tissue from FAD patients, different AD mouse models and extracellular beta-amyloid to show the effect of Beta-amyloid in mitochondria-ER contacts (MERCS); and how this affects autophagosome formation.

Authors show that B-amyloid increases MERCS and that this is associated with a decrease in Mfn2. In addition, APP neurons show an earlier autophagosome formation than control neurons after starvation which is preceded by an increase in MERCS and in several mitochondrial bioenergetics parameters (MMP, ATP and respiration) in both cases.

This is a well written and technically sound study. Although some results appear confusing, such as the variations in MERCS parameters within the different mice ages (Figs 2, S2), experiments in the neuronal cultures show clear results.

Comments:

Results would be more complete if changes in mitochondrial calcium were assessed, to functionally link the increased MERCS and the changes in the mitochondrial bioenergetics parameters.

It is not clear for how long was synthetic beta-amyloid applied to the cells before the measurements took place. Is this an acute effect?

How do authors interpret the lack of changes in p62 levels in APP in opposition to control?

According to these results, in the basal conditions tested, beta-amyloid has a positive impact in mitochondrial bioenergetics as shown by the increased ATP production among others (Fig S5). Please discuss these findings in contrast with many other publications showing (opposite) bioenergetics deficits induced by Abeta, such as mitochondrial depolarization, impaired respiration and reduced ATP levels.

Author Response

1.The comment is highly relevant, however currently we do not have the method to measure mitochondrial calcium established in our laboratory and therefore such data cannot be included in this revision. However, the topic is addressed in the manuscript and the role of calcium in MERCS and neurodegeneration is for exampled discussed on line 553-559.

2.By lapse, the incubation time with the synthetic Aβ was missing in the Material and Methods. We thank the reviewer for noticing this. We have now added this incubation time in line 164.

3. Impairment of autophagy normal function in AD models has been described, with accumulation of autophagic vacuoles and impaired degradation of its content. This could explain the lack of changes in our AD mouse model (AppNL-F) during starvation where p62 would accumulate in autophagic vacuoles and not being degraded. Another explanation could be p62-cargo mistargeting due to the increased Aβ-levels. This is explained on line 389-391.

4.The reviewer is correct about this apparent discrepancy of results. In two almost completed studies from our laboratory we show that mitochondrial functions are increased in young AppNL-F mice and that the mitochondrial functions then deteriorate as the pathology develops in the aging mice (Dentoni et al. and Naia et al., manuscripts in preparation). These data are in-line with results shown here with the positive impact in mitochondrial bioenergetics detected in primary neurons derived from AppNL-F mice. Thus, we hypothesize that in early disease stages neurons respond to increased Aβ levels by upregulating mitochondrial functions as an attempt to protect from the cellular stress that Aβ induces. As pathology develops with further Aβ generation and accumulation, the mitochondria cannot cope and Aβ becomes toxic to mitochondria.

Reviewer 3 Report

This manuscript evaluated the interplay between mitochondria-ER contact sites (MERCS),mitochondria structure/function and autophagy in three AD models, and in primary neurons derived from those models. Electron microscopy were used for identifying mitochondria ultrastructure and Seahorse analysis were used for mitochondria function. In summary, authors reported that amyloid beta can affect cell homeostasis by modulating MERCS and consequently altering mitochondria activity and autophagosome formation. 

Comments:

  1. Line 289-305, should be part of the figure legend. Please adjust the fonts and size accordingly.
  2. Line 383, since Bafilomycin is am inhibitor of the fusion between autophagosomes and lysosomes, please provide explanation how the  "fusion is not impaired" is concluded from Fig S4G experiment after addition of Baf.
  3. Fig 4D/E, seahorse experiments are presented as %, please present as raw numbers (pmol/min/ug protein). And it would be nice to run WT cells and APP cells on the same plate.
  4. Mfn2 was found to be decreased in AD patients and AD mice. However, Mfn2 levels are increased after starvation. Please provide discussion.

Author Response

1.We thank the reviewer for this observation. We have adjusted this accordingly (line 295-311).

2. Baf inhibits the fusion between autophagosomes and lysosomes, preventing LC3-II from being degraded and, therefore, remains associated in autophagosome instead of being cleared. In Fig S4G, when we incubate PCN (WT or AppNL-F) with Baf during starvation, we see an increase of LC3-II, when compared to PCN non-Baf treated cells and at the same starvation time point. This shows that fusion between autophagosomes and lysosomes was taking place in non-Baf treated cells, otherwise no effect of Baf (detected as LC3-II accumulation) levels would have been detected. This was changed in the manuscript accordingly in order better explain this reasoning (line 392-398).

3.Primary cortical neurons are prepared from a pool of different embryos and, therefore, variations are expected. To control for this variability between preparations we chose to express the Seahorse data as % of control and we would like to keep it this way.

Regarding the comparison between the WT and APP in the same plate that was done, and it is presented in Fig S5A and discussed in line 457-461.

4. Both data sets support the hypothesis that Mfn2 is a negative regulator of MERCS (lines 547-549 and 574-578 of the discussion). Decreased levels of Mfn2 in SAD and AppNL-F PCN leads to an increase of MERCS (i e more contact between mitochondria and ER) in these models, on the contrary the increase of Mfn2 in starved wt PCN leads to decrease in MERCS (i e less contact between mitochondria and ER) during autophagosome formation.

Round 2

Reviewer 1 Report

The authors fulfilled all my comments